# The Role of Estrogens and Vitamin D in Cardiomyocyte Protection: A Female Perspective

**DOI:** 10.3390/biom11121815

**Published:** 2021-12-02

**Authors:** Clara Crescioli

**Affiliations:** Department of Movement, Human and Health Sciences, Section of Health Sciences, University of Rome “Foro Italico”, 00135 Rome, Italy; clara.crescioli@uniroma4.it; Tel./Fax: +39-06-36733395

**Keywords:** estrogens, vitamin D, cardiomyocyte remodeling, cardioprotection, women

## Abstract

Women experience a dramatical raise in cardiovascular events after menopause. The decline in estrogens is pointed to as the major responsible trigger for the increased risk of cardiovascular disease (CVD). Indeed, the menopausal transition associates with heart macro-remodeling, which results from a fine-tuned cell micro-remodeling. The remodeling of cardiomyocytes is a biomolecular response to several physiologic and pathologic stimuli, allowing healthy adaptation in normal conditions or maladaptation in an unfavorable environment, ending in organ architecture disarray. Estrogens largely impinge on cardiomyocyte remodeling, but they cannot fully explain the sex-dimorphism of CVD risk. Albeit cell remodeling and adaptation are under multifactorial regulation, vitamin D emerges to exert significant protective effects, controlling some intracellular paths, often shared with estrogen signaling. In post-menopause, the unfavorable association of hypoestrogenism-D hypovitaminosis may converge towards maladaptive remodeling and contribute to increased CVD risk. The aim of this review is to overview the role of estrogens and vitamin D in female cardiac health, speculating on their potential synergistic effect in cardiomyocyte remodeling, an issue that is not yet fully explored. Further learning the crosstalk between these two steroids in the biomolecular orchestration of cardiac cell fate during adaptation may help the translational approach to future cardioprotective strategies for women health.

## 1. Introduction

Sex-related differences consistently contribute to the clinical heterogeneity characterizing aging in respect to cardiovascular diseases (CVD), which currently represent the leading cause of illness and death in the Western world [1]. Indeed, women show a higher prevalence of age-related cardiac defects, including left ventricular hypertrophy, increased end-diastolic pressure, diastolic dysfunction, fibrosis, inflammation, oxidative stress and lower exercise capacity [2].

The midlife estrogen withdrawal is recognized as being the main traditional cause of CVD increase and heart failure (HF) in post-menopausal women [3]. Indeed, the menopausal transition associates with adverse organ macro-remodeling, which is in turn ascribable to cardiac and endothelial cell micro-remodeling (which occurs in a sex-specific mode [4,5]). Those effects largely depend on the presence of estrogen receptors (ER) within the myocardium and endothelium, and encompass many cellular biological functions by genomic and non-genomic mechanisms [6,7,8]. Nevertheless, estrogens/ER alone are unlikely to entirely explain heart disease presentations and outcomes in women, so that the identification of bio-factors contributing to the higher risk in females is still subject to ongoing debate.

Low vitamin D can critically promote molecular alterations toward an aberrant cell remodeling. Indeed, vitamin D deficiency increases the risk of CVD development, impacting cell morphology, metabolism and function [9,10]. Upon binding with its specific receptor VDR present in vascular and cardiac cells, vitamin D affects several biomolecular and cellular processes. As for estrogens, whereas vitamin D actions onto vascular cells are quite exhaustively covered in literature, its molecular effect onto cardiomyocytes is still incompletely understood, especially from sex-dependent standpoint. Yet, an aberrant remodeling of cardiomyocytes, as occurring in unfavorable environment, is pointed to as the main trigger of a compromised cardiac function [11,12]. Conversely, in normal conditions, cardiac cell remodeling allows adaptive responses and favors cardio-protection. Women after menopause tend to have pronounced hypovitaminosis D, which, along with hypoestrogenism, seems to permit more relevant negative effects on heart health.

This review aims to offer first an overview on the role of estrogens and vitamin D in cardiovascular female health, focusing on cardiomyocyte remodeling. Then, the potential synergistic effect of low estrogens-low vitamin D combination, both hormone-deficient conditions typically occurring in postmenopausal life, is speculated. Brief comments on the impact of possible protective interventions, such as a combined supplementation of these hormones or physical activity are mentioned in the conclusive part.

## 2. Cardiovascular Health: A Sex Hormone Matter

Although sex-dependent differences in cardiac aging and CVD development are multi-factorial, there is evidence pointing out the role of estrogens/ERs. Indeed, the drastic difference between aging women and men is unquestionably related to the rapid decline of female sex hormones, associated with menses ending. Female heart is known to better resist different insults, and, accordingly, women are better protected from heart diseases compared to men. This remarkable “female advantage” is lost with menopausal transition or metabolic diseases such as diabetes, which significantly increased the incidence of cardiovascular morbidity and mortality in post-fertile women life, as summarized elsewhere [13,14]. Indeed, some functional and structural sex-dependent intrinsic differences, including heart contractility and rhythm, which affect contraction and relaxation, less left ventricular mass, lower chamber volume, thinner wall thickness or electro-mechanical function as seen in females, might explain at least in part, the sex dimorphism in clinical manifestations and responses to treatments. In general, women present less severe clinical symptoms, but the prognosis after heart attack is worse, with about 40% mortality in women vs. 25% in men within a year [14]. Women are more frequently affected by hypertension (which likely contributes to the different left ventricular dysfunction and arterial stiffness) and are more prone to cardiometabolic syndrome (which exacerbates sex-related bias in cardiac disease occurrence) and disease sequelae, as addressed later in this manuscript. In particular, left ventricular diastolic dysfunction, the typical clinical presentation of HF with preserved ejection fraction (HFpEF) predominant in women, seems to be facilitated by estrogen deficiency, affecting intra-cellular calcium homeostasis, cyto-skeleton and extra-cellular matrix rearrangements [15].

From the pioneering studies, the role of estrogens in cardioprotection has been essentially related to the arterial vasodilation through direct genomic and non-genomic actions onto vascular cells [16,17,18,19].

In endothelial and vascular smooth muscle cells, in fact, estrogens—especially 17β-estradiol/E2, the predominant biologically active form—upon binding with α and β subtype ER (ERα and ERβ) and with G-protein coupled estrogen receptor (GPER), activate a cascade of intra-cellular signaling paths, such as phosphoinositide 3-kinase-serin/threonine-specific kinase B (PI3K/Akt)/endothelial nitric oxide synthase (eNOS) and mitogen-activated protein kinases (MAPK)/eNOS, allowing nitric oxide (NO) release, vascular relaxation and vasodilation [20,21,22]. These effects converge towards the regulation of the vascular tone against hypertension and protect from high pressure-induced damage of arteries and atherosclerosis [23]. In addition, atherosclerosis prevention also depends on the anti-oxidant action of estrogens, which reduces the deposition of circulating cholesterol in arteries wall and limits inflammation [24]. With the menopausal transition, estrogen-induced protection is lost, as endogenous hormone concentration in women drops to the small amount produced in extra-gonadal sites (adrenal cortex cells, aortic smooth muscle cells, adipose tissue, brain, bone), similarly to ovariectomized women or men [25]. The scheme in Figure 1 summarizes the main effects of estrogens in vascular and cardiac cells.

### Estrogens in Males, Androgens in Females

The conversion of testosterone to estrogens by aromatase seems to exert some protecting effects in males as well [26]. In fact, men with E2 deficiency or E2 resistance, due to specific mutations in the cytochrome P450 aromatase gene (Cyp19a1) or in the ERα gene (ESR1), respectively, show an increased risk of CDV in association with total cholesterol level rise, insulin resistance (IR) and type 2 diabetes (T2D) development, defects in glucose tolerance and vasodilation [27,28,29,30,31]. Treatment with estrogens can normalize the cardiac function in male mice with HF, induced by aromatase activity suppression [32]; in line with this experimental observation, a reduced risk of CVD events from endogenous estrogens is reported in elder men [33].

To date, whereas in men the exact role of E2 with regard to heart function remains questionable, estrogens seem undeniably protective in women, considering the CVD risk before and after menopause when disease incidence becomes equal or even greater than that one observed in men-making CVD the leading cause of death in both sexes [26,34,35].

Nevertheless, it should be recalled that long-term treatment with estrogens (especially synthetic drugs), as contraceptives or hormone replacement therapy (HRT), is associated with super-oxide radical accumulation, inflammation and hypertension [36,37], processes that can worsen cardiac myopathic changes. According to the “theory of timing and opportunity”—based on menopausal stage and time of hormone administration—the cardiovascular benefit seems limited to younger women, who initiated HRT in early peri-menopausal stage [21,38,39,40]. Particular attention should be given to HRT in women with a condition known as metabolic or cardiometabolic syndrome, a cluster of diseases, including obesity, dyslipidemia, hypertension and IR, which increases with menopause (present in 40% of post-menopausal women) and shows some sex dimorphism as well. In fact, although this condition represents a primary risk factor for diabetes and CVD in both sexes, it is speculated that it contributes to the different cardiovascular sequelae in men and women. Whereas HRT confers beneficial effects on metabolism, acting onto abdominal fat, blood lipid, adipokine profile, vascular resistance and oxidative stress, it does not provide sufficient cardio-protection in women with pre-existing diabetes, CVD or related risks [13,14]. This discrepancy might be partially due to estrogen bio-molecular interactions with the cardiovascular system, leading to differences in response to the treatment [41]. Further studies are mandatory to fully explain the underlying mechanism(s). Meanwhile, given the existing controversy on HRT pros and cons, a careful evaluation of potential real benefits is recommended when considering this therapeutic strategy.

The cardiovascular health in females seems under androgen control as well. The excess of androgens produced by ovary in menopause, not balanced by estrogen production, is hypothesized to negatively affect women cardiovascular function and to increase the cardiovascular risk in association with T2D and diabetic cardiomyopathy [13,42]. The link between hypoestrogenism, hyper-androgenism and cardiometabolic risk in women is still a challenging issue and surely deserves further studies [43,44]. Nevertheless, it is undeniable that estrogens broadly impact women’s cardiovascular health through rapid and genomic mechanisms exerted onto endothelium and vascular cells, as exhaustively covered in a recent report [19] and not discussed in this review. Herein, the attention is on some significant direct effects of estrogens onto cardiac cell remodeling, which represents the critical event driving toward health maintenance or disease development.

The physiological remodeling of the cardiomyocyte, both in males and females, is under control of different factors finely orchestrated to allow compensatory functional adaptation in response to various stimuli, i.e., aging, stress, physical exercise or pathological challenges. If, for any reason, an adaptive cell remodeling is not properly maintained, maladaptive processes take place, and, consequently, cardiac function is not safeguarded.

## 3. Estrogens and Cardiomyocyte Remodeling

The protective effect of estrogens on female heart against various stress challenges, such as hypertrophic, ischemic or cytotoxic stimuli, involves direct actions of these hormones in the cardiomyocyte. The sex difference underlying this process undoubtedly includes multi-factorial reasons, but the major evidence points to a causal role of the sex steroid hormone E2 and its receptors (ER) in the physiology and pathophysiology of the heart. Interestingly, key events like cardiac calcium (Ca^2+^) ion channel activity and mitochondrial function are regulated in a sex-specific manner, as discussed below.

### 3.1. Estrogen Receptors

The presence of α and β cardiac ERs in ventricular and atrial cells of adult and neonatal heart is known since quite ago, and successively confirmed in female and male mice [45,46,47,48]. ER subtypes are present in cardiac cell cytosol with different subcellular localization (being ERα located in or adjacent to plasma membrane) and exert translational and post-translational regulatory mechanisms [48,49,50]. Studies in animals carrying cardiomyocyte-specific deletion or overexpression of ERα (ERKO-mice or csERα-OE mice, respectively) documented that this receptor subtype leads female cardiomyocytes to more efficient recover after cardiac injury, albeit it is dedicated to heart mass regulation in both sexes [50,51,52,53].

ERβ dysfunction is linked to cardiomyocyte disarray and important alterations in tissue architecture, including nuclear structures and gap junctions [54]. Of note, females, not males, lacking ERβ show a significantly reduced post-ischemic cardiac recovery [55,56]. These findings support an ERβ-dependent protective role after cardiac injury in females. In presence of cardiomyocyte-specific ERβ over-expression (csERβ-OE mice), differences between sexes were present neither in basal cardiac morphology/function/weight, nor in recovery/survival after cardiac injury [54]. Female and male mice, indeed, showed the same improvements in several cardiac parameters, except for left ventricular volume and ejection fraction, being both more pronounced in males. This effect likely depends on a reduced cardiomyocyte remodeling towards fibrosis, as observed in males. The main protective action of ERβ seems to rely on a better protection of the sarcoplasmic/endoplasmic reticulum Ca^2+^-ATPase 2a (SERCA2a) system and Ca^2+^ reuptake post-injury.

In addition to ER α and β, cardiomyocytes express GPER, a membrane receptor essentially mediating non-genomic rapid actions [57]. Only male GPER-knock out (KO)-mice seem to develop impairments in cardiac function. Defects in heart structure and function observed in cardiomyocyte-specific GPER-KO animals were exacerbated in aging males [58]. This difference observed between males and females likely mirrors the difference found in gene expression profile related to cardiomyocyte GPER-deficiency and sex, as mitochondrial genes were enriched only in female GPER-deleted cardiomyocytes vs. wild type [58,59].

Of note, mitochondria seem to play one of the major roles in orchestrating biomolecular events in cardiac cell remodeling and function.

### 3.2. Sex-Dimorphic Mitochondrial Function

A marked sexual dimorphism is reflected in different mitochondrial calcium handling, higher oxidative capacities, and greater resistance to oxidative stress. Mitochondria content from female cardiomyocytes seems lower but with higher efficiency and differentiation levels, as compared to male ones [60]. Mitochondria from cardiac cells express both ER subtypes, which contribute to ER-related regulation of these organelles by affecting contractility (involving ATP supply), Ca^2+^ homeostasis, reactive oxygen species (ROS) formation and cell apoptosis [61,62,63]. E2 are reported to enhance mitochondrial respiration and reduce ROS, both processes associated with a lower incidence of CVD in women before menopause; accordingly, during aging, estrogen decline is associated with mitochondrial damage, tissue/cell loss of function and increased risk of disease, as exhaustively described elsewhere [60]. Estrogens and ER play a pivotal role also in the regulation of Ca^2+^ ion channel signaling and contractility. Sex difference in contracting function, i.e., excitation–contraction (EC) coupling, involving cardiac L-type channels, are observed in humans and animals. Human ventricular cardiomyocytes of female failing heart retain greater contractility and enhanced L-type Ca^2+^ current, as compared to men [64,65,66]. So far, estrogen decline is largely engaged in Ca^2+^ signaling deregulation and mitochondria defective functioning, the two main mechanisms pointed to as the most important events involved in aging heart and CVD development/progression [63,67,68].

A study in ovariectomized mice documents the importance of AMP-activated protein kinase (AMPK) in estrogen-mediated cardio-protection via intra-cellular Ca^2+^ and cell contractility regulation [69]. This signaling molecule involved in energy metabolism and cardiac function regulation is documented to be permissive for estrogen-mediated maintenance of cardiac homeostasis, and, noticeably, can restore a correct cardiac glucose transport, impaired after ovariectomy. The effect of estrogens in cardiomyocytes are schematized in Figure 1.

Hence, the role of estrogens onto these cellular processes within cardiac myocytes is unquestionable; nevertheless, it should be underlined once more that sex hormones are not the unique steroids controlling cardiomyocyte function. In particular, the following paragraph will describe how cardiovascular health and cardiac cell remodeling depend on vitamin D, another steroid hormone exerting important biological actions, beyond its classical skeletal effects. The main functions mediated by ER different subtypes and VDR are summarized in Table 1.

## 4. Cardiovascular Health beyond Sex Hormones: Vitamin D Matters

The effect of vitamin D on cardiac aging, function and disease still represents a hot issue in research. Since quite a long time ago, vitamin D is known to encompass a wide spectrum of biological actions, which significantly affect cardiovascular homeostasis, as suggested by the clinical association between vitamin D deficiency and different cardiovascular events [77,78,79].

It should be kept in mind that vitamin D is a molecule with quite a special identity, presenting features typical of a nutrient, a hormone and a rapid regulating factor; it is able to affect human health through a fine-tuned regulation of cell functions, by actively participating in biomolecule networking, as recently reported [80,81,82].

Concerning basal vitamin D level, males seem to retain higher serum hormones than females (likely due to body composition [83]), albeit some controversial data exist, maybe depending on several variables. Data from cross-sectional studies in more than 2000 Norwegian morbidly obese subjects and in about 4000 Indian obese and diabetic patients document higher odds of vitamin D deficiency in men, likely related to abdominal adiposity, and higher cardiometabolic risk in association with lower vitamin D, respectively [84,85]. Conversely, studies in coronary artery disease report that lower vitamin D levels, as found in women associate with disease severity [86]. However, rather than absolute differences between male and female vitamin D basal levels, some sex-specific determinants of vitamin D status (i.e., body fat, presence/absence of pathologies, exposure to sun, diet and hormone supplementation and sedentary lifestyle) should be considered, especially in scenarios evaluating preventive strategies in diseases with a strong female bias, as shown, i.e., in auto-immune diseases [87,88,89].

Despite data controversy on sex-related difference in basal vitamin D and unsubstantiated remarks regarding D level and heart health in females and males, the need to improve vitamin D status related to cardiovascular health in the general population is unequivocally recognized [90].

From previous human and experimental studies, the last ones performed in restriction diet models, vitamin D deficiency emerges to associate with increased arterial blood pressure, vascular oxidative stress, modifications in cardiac gene expression, left ventricular hypertrophy, cardiac inflammation, coronary artery disease severity, fibrosis and apoptosis [78,91,92]. In humans, low serum vitamin D level associates with impairments of left ventricular structure and function [93,94]. The scheme in Figure 2 summarizes some of the main detrimental effects of vitamin D deficiency on CVD.

Hypovitaminosis D enables several cardiovascular alterations associated with CVD development and severity (CAD: coronary artery disease).

Indeed, vitamin D, upon binding VDR, can control cardiovascular homeostasis by affecting a variety of mechanisms, including cellular proliferation/hypertrophy, blood pressure and renin–angiotensin system. Clinical evidence indicates that vitamin D, like estrogens, directly impacts on vasculature and endothelial cells, as shown by the positive correlation between vitamin D level and arterial compliance, improvements in endothelial function, a reduction in vascular fibrosis in response to injury and a decrease in those inflammatory cytokines underlying HF development [79,95,96,97,98,99]. To date, concerning cardiovascular disease prevention based on endothelial function protection, the beneficial effects of vitamin D on the vasculature are still under debate, especially in light of the possible dose-related vasculature calcification during hormone supplementation. This specific issue is beyond the aim of this review and is plenty covered elsewhere [100,101,102].

Conversely, vitamin D actions directed onto the cardiomyocyte are less covered in literature and, understandably, the main reports are on animal models, which not always can be translated to humans. Another gap (even greater) in literature concerns possible sex-specific associations between vitamin D deficiency and CVD. Those topics are addressed in the following paragraphs.

## 5. Vitamin D and Cardiomyocyte Remodeling

Nowadays, it is widely recognized that vitamin D level affects cardiovascular health and cardiac cell adaptation, impinging on several intra-cellular processes, including Ca^2+^-dependent mechanisms such as Ca^2+^-binding protein synthesis, adenylate cyclase activation, voltage-dependent Ca^2+^ channel rapid activation and sarcoplasmic reticulum Ca^2+^ uptake and release [70]. Although the heart is not considered a traditional target tissue of vitamin D, functional vitamin D receptors (VDR) are present in human and animal cardiac myocytes and exert some biological protective actions [103,104] (please see Table 1).

Some investigations suggest the importance of vitamin D/VDR system in controlling cardiac hypertrophy, a dominant feature of several heart disease, and diastolic function.

I.e., cardiomyocyte-specific VDR deleted mice (VDRKO mice) and VDR-deleted rats exhibit ventricular hypertrophy, increased matrix turnover and kinetics alteration [71,105,106]. In those experimental models, liganded VDR works against hypertrophy by counteracting the pro-hypertrophic calcineurin/nuclear factor of activated T-cells (NFAT) and modulatory calcineurin inhibitory protein 1 (MCIP 1), and ameliorates contractility and relaxation kinetics, respectively [70,71,72]. Another example of heart cell protection upon VDR activation comes from a model of diabetic fatty rats (Zucker), in which the treatment with vitamin D can limit cardiomyocyte autophagic activity and damage through the inhibition of FoxO1 translocation and transcriptional activity [73]—the same mechanisms described in osteoblasts (the classical cell target of this hormone) [107].

In human cultured cardiomyocytes exposed to maximal pro-inflammatory challenge, a VDR agonist blunts the intra-cellular activation of signal transducer and activator of transcription 1 (Stat1), induced by interferon (IFN)γ, and notably, almost prevents phosphorylation/nuclear translocation of nuclear factor-kB (NF-kB), induced by tumor necrosis (TNF)α [74]. The latter effect seems particularly intriguing, considering the relevant role played by this prototypic inflammatory cytokine in adverse cardiac remodeling toward heart failure and disease outcome [108,109].

Deficits in VDR expression and low vitamin D are shown to allow detrimental alterations in metabolism, signaling and ionic currents of cardiomyocytes. As an example, in presence of vitamin D insufficiency, ROS generation is enhanced, and, in turn, promotes a cascade of pro-hypertrophic intracellular signaling, i.e., MAP kinase cascade, extra-cellular signal-regulated kinase 1/2 or ERK 1/2, ERK 5, and NFκ-B, c-Jun NH2-terminal kinase 1/2 (JNK), p38 mitogen-activated protein kinase [110], all converging towards fibrosis development. The length of vitamin D deficiency seems to determine the intensity of cardiomyocyte alterations, and, noticeably, the restoration of adequate vitamin D level can protect cardiomyocytes from aberrant signaling [111].

Membrane-bound VDR is shown to mediate rapid non-genomic processes and control the contraction of cardiomyocyte sarcomere, through caveolin 3 interaction, an integrated mechanism also reported in other cells [112,113,114,115].

Non-genomic VDR activation is also described to regulate post-translational events through epigenetic effects by the generation of specific microRNA (miRNA) [116,117]. Of note, miRNA gene regulation by VDR may represent an important mechanism involved in the control of signal transduction through the recognition and degradation of target mRNAs level and translated proteins [118]. Indeed, the miRNA regulatory network may largely affect signaling molecules that exert pleiotropic effects all through the body, such as vitamin D. In fact, since miRNA targets may number up to hundred, it is conceivable that this mechanism can largely amplify the pleiotropic effects of vitamin D/VDR in cardiac cells, as occurs in other tissue and cells, i.e., adipose, cancer or bone cells, striated muscle cells, in which a mutual interaction is described [119,120,121,122,123,124,125]. Figure 3 schematically reports some of the main regulatory actions of vitamin D in cardiomyocytes.

### Vitamin D and Sex-Dimorphic Cardiac Metabolic Flexibility

Another cardioprotective mechanism of vitamin D is the regulation of mitochondrial metabolism and energy supply [75].

The energetic regulation of the cardiomyocyte is a quite complex process characterized by substrate promiscuity (fatty acids, carbohydrates, amino acids, lactates and ketons) to allow for multiple substrate utilization for energy production, in view of the high energy demand. Cardiac cells predominantly utilize fatty acids and rapidly shift to other substrates as they become abundantly available, to warranty an adequate ATP supply [126]. This process, known as metabolic substrate flexibility, permits the cell to adapt in response to physiologic conditions (i.e., during exercise) or pathologic challenge, such as hyperglycemia or inflammation [12]. Cardiac energetics and metabolic substrate flexibility display some degree of sex-dimorphism as well. As an example, according to human in vivo studies in healthy young adults, a women’s heart appears to utilize more oxygen and less glucose as compared with the heart of age-matched man [76]. This effect is dependent, in part, on estrogen-induced eNOS upregulation, which decreases glucose transporter (GLUT)-4 translocation to cell surface and, in turn, reduces glucose uptake/utilization by the cardiomyocyte [127,128]. Thus, female cardiac cell energetics depends more on fatty acid oxidation, which requires more oxygen consumption. The well-perfused oxygenation is one of the mechanisms underlying cardio-protection in healthy female heart. However, recent studies on exercise-induced remodeling in males and females show that female cardiac cells likely retain a lesser degree of metabolic flexibility to adaptation in stress or disease conditions [129]. Noticeably, in cardiac cells vitamin D is reported to impinge on the energy substrate balance, regulating fat uptake/fatty acid β-oxidation via sirtuin 3 [75].

In this scenario, we could figure the cardiomyocyte as the cellular crossroad where the hormonal signaling from estrogens and vitamin D may meet and intersect to regulate cell remodeling and drive cardiac function towards adaptation or, in case of simultaneous hormone deficiency, maladaptation, as depicted in Figure 4.

To date, the research on this specific topic is still in its infancy, but the cooperation between vitamin D and estrogen in the regulation of some other important biological functions may provide mechanisms and models as examples.

## 6. Vitamin D and Estrogen Cooperation: Learning from Examples

The classical example of interaction between vitamin D and estrogens comes from bone health in women: both hormones act in synergy to promote osteoblast proliferation and differentiation, in association with E2-induced VDR upregulation, via MAPK, ERK1/2 signaling, as well documented in past studies [130,131,132].

The cooperation between vitamin D and estrogens in mice striated muscle cells results in a potentiated induction of cAMP response element binding (CREB) phosphorylation and c-Fos protein expression, via MAPK and ERK-dependent intracellular cascade activation [133]; this observation discloses hypothesis on fast non-transcriptional responses evoked by hormone combination to promote muscle recovery and function.

Another example of vitamin D and estrogen signal cooperation is elegantly shown by human and experimental studies in autoimmune multiple sclerosis (MS), a disease with higher female bias and marked hypovitaminosis D. In mice with autoimmune encephalomyelitis (EAE, resembling human MS), only intact females can achieve protective effects via vitamin D [134,135]. After estrogen implants, EAE ovariectomized females fully retrieve vitamin D-induced beneficial effects. The protective effect relies on an enhanced E2 biosynthesis promoted by vitamin D and an increased VDR expression induced by E2, allowing the two hormones to act together toward disease remission [135]. This vitamin D-mediated protection is female specific and is observed neither in ovariectomized females nor in males. Of interest, similar results come from a human large prospective study in MS subjects: the higher level of vitamin D naturally occurring in summer inversely associates with MS incidence and MS disability only in women, suggesting that estrogens, somehow, are selectively permissive for the beneficial effects of vitamin D [136].

The importance of vitamin D-estrogens mutual crosstalk in health/disease discrimination is not so surprising when thinking of some pioneering investigations in women, which showed that vitamin D level is higher in high estrogen conditions (i.e., pregnancy, ovulation, HRT in post-menopause) [137,138]. Hypovitaminosis D and low free testosterone associate with particularly adverse clinical outcome in men referred for coronary angiography, suggesting some interplay of these hormones in males [139]. Low vitamin D and testosterone deficiency are considered typical features in men with advanced HF; however, vitamin D supplementation in this group of patients cannot prevent the decline in testosterone indices [140].

Concerning the specific topic on female heart function, a quite recent study postulates a synergistic role of vitamin D and E2 in postmenopausal women with metabolic syndrome, a cluster of simultaneous cardiovascular risk factors, leading to an increased risk of heart disease, stroke and T2D, as previously mentioned [141]. The study shows that low vitamin D likely increases the risk of disease in women with hypoestrogenism, documenting a stronger inverse correlation E2-metabolic syndrome in women with D hypovitaminosis vs. women with a normal vitamin D level. This observation is in line with a previous study performed in African American women, who are most vulnerable than other age-matched race or ethnic groups because of their higher risk for this disease, likely related to further reduction in estrogen and vitamin D levels [142].

Although these papers are far from confirming that a low estrogen level favors the disease via vitamin D deficiency, some mechanisms are conceivable, involving hormone-dependent merged actions onto endothelial cells and cardiomyocytes. Meanwhile, in vascular (endothelial and smooth muscle) cells, vitamin D and estrogens are known to interdependently regulate blood pressure by the release of potent vasodilators, such as NO, prostaglandins (PGs), nitric oxide and calcitonin gene related peptide (CGRP) [142,143], the potential cooperating mechanism(s) of these hormones in cardiomyocytes are scarcely described. However, as addressed before in this review, it is well recognized that vitamin D can directly affect signal transduction mediators and ion channels in cardiomyocytes, often sharing the same biomolecular signaling with E2, such as NO, eNOS, CGRP and peroxisome proliferator activated receptor (PPAR)α—the latter one is involved in lipid and glucose metabolism regulation [142]. Thus far, the recognition that cardiomyocytes express all the isoforms of nitric oxide synthase (NOS), the three isoforms of PPAR, CGRP and progesterone (PG) receptors [144,145] can hopefully open further hypothesis and perspectives on possible synergic mechanisms in these cells, whose structure and function likely respond to vitamin D-other hormone interplay.

## 7. Conclusions

Although sex undeniably matters in cardiac health, vitamin D tightly affects heart function in females and males. The detrimental convergence of hypovitaminosis D and estrogen deficiency, naturally occurring with menses ending, reveals the higher vulnerability of postmenopausal women experiencing this condition, associated, indeed, with a significant increase in CVD rate vs. age-matched men. Estrogens, the molecules classically considered as the main responsible for the sex-dimorphism in cardiac function, unlikely can fully cover this difference. Vitamin D, like estrogens, finely impinges on heart remodeling in response to different challenges (i.e., aging, volume or pressure overload, exercise, necrosis). Nowadays, cardiomyocyte remodeling, the cellular event driving whole organ macro-remodeling (via adaptive or maladaptive responses) is recognized to be sex-dimorphic and affected by vitamin D. In this scenario, supplementation with both hormones apparently would offer a promising approach in women CVD prevention, especially in post-menopause life. However, beside the limit of HRT (useful only when taken within a limited timeframe), data on the efficacy of vitamin D supplementation in CVD prevention are contradictory [146,147,148]. In the U-shaped relationship between vitamin D and cardiovascular risk, 20 ng/mL is considered the vitamin D serum level associated with an apparent minimum risk, and a dose > 4000 IU/day as supplement seems necessary to affect heart remodeling in vitamin D deficient subjects with HF [90,149,150]. Indeed, general important concerns still exist in several aspects, including the lack of a clear indication of the optimal dose requirement to respond to extra-skeletal needs, or the lack of a clear definition of vitamin D insufficiency/deficiency, based on well-defined serum ranges (still missing as well) [151,152,153]. Data from trials on D hormone supplementation, either alone or combined, are unsatisfactory likely due to the high variability in protocols and heterogeneity of the studied populations, since general health status, ethnicity, sedentary habit are often not defined, as recently summarized elsewhere [80,81]. Furthermore, recent evidence highlights the major role of exercise-induced sex-dependent heart remodeling during life of men and women at different ages, either sedentary or physically active [129]. This topic is not addressed in the present review, both for length limits and because, in our opinion, it would deserve a dedicated issue.

Thus far, albeit epidemiological studies and metanalyses report an unequivocal association between vitamin D status and heart remodeling, there are still inconclusive remarks from the sex-specific standpoint [90,150]. Research on this specific field is still in its infancy and we are aware of the difficulty to consider, present and discuss hypovitaminosis D and sex-related CVD as not separate but interconnected issues, within more complex scenario(s).

Herein, the interest is confined and focused on the importance of vitamin D-estrogen interplay in cardiomyocyte adaptation, to drive the attention on an issue still not fully explored and, maybe, to be hypothesis generating.

While waiting for mandatory well-designed trials to overcome the existing bias, further progress of basic research on estrogen–vitamin D crosstalk and their orchestration of the cardiomyocyte remodeling would represent an important step forward to narrow the large gap in the knowledge of this topic, in consideration of the translational approach to future strategies in women health prevention and therapy.

## Figures and Tables

**Figure 1 biomolecules-11-01815-f001:**
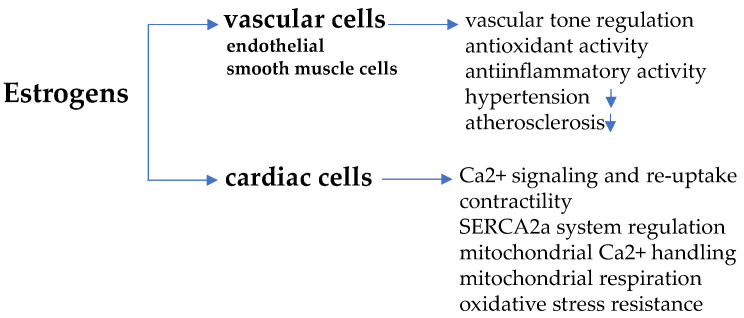
Estrogen-induced regulation in vascular and cardiac cells. Estrogens work against hypertension and atherosclerosis through anti-oxidant/anti-inflammatory activity and regulate vascular tone, acting on endothelial and smooth muscle cells. In cardiomyocytes, estrogens regulate cell contractility affecting Ca^2+^ dependent signaling, mitochondria function and sarcoplasmic/endoplasmic reticulum Ca^2+^-ATPase 2a (SERCA2a) system.

**Figure 2 biomolecules-11-01815-f002:**
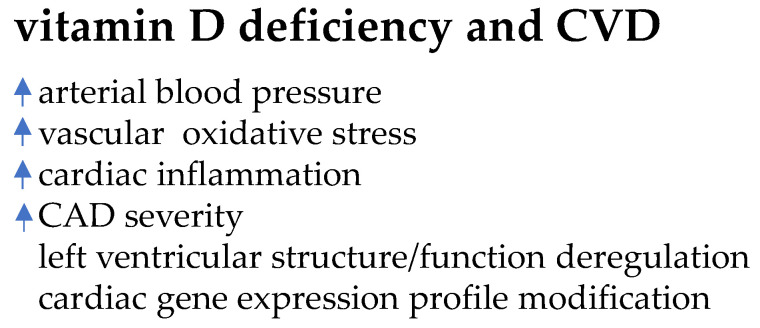
Vitamin D deficiency-induced effects on cardiovascular system.

**Figure 3 biomolecules-11-01815-f003:**
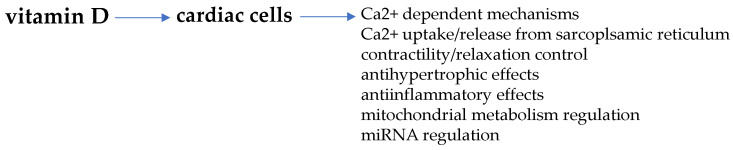
Vitamin D-mediated regulation of cardiac cells. Some important cardiomyocyte functions regulated by vitamin D are targeted by estrogens as well.

**Figure 4 biomolecules-11-01815-f004:**
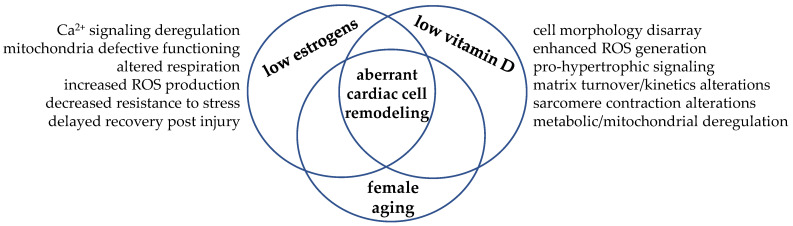
Hypoestrogenism and hypovitaminosis D may intersect in the cardiomyocyte and drive aberrant remodeling. The simultaneous low level of these steroid molecules—a typical hormonal condition experienced post menopause—allows biomolecular aberrant signaling and cell maladaptative response. As these hormones often share the same biomolecular signaling, the hypothesis of interdependent synergic actions within the cardiomyocyte has taken place.

**Table 1 biomolecules-11-01815-t001:** Estrogen receptor (ER) and vitamin D receptor (VDR) in cardiomyocytes. The table summarizes the main intra-cellular effects mediated by ER sub-types and VDR, affecting cardiomyocyte remodeling.

Receptor Type	Mediated Function	Refs.
ERα	growth control post-injury recovery (more efficient in females)	[50,51]
ERβ	cell/tissue structure, gap junction post-ischemic recovery (only in females) anti-fibrotic activity (more efficient in males)	[54,55,56]
ERα/ERβ (in mitochondria)	mitochondria sex-dimorphism Ca^2+^ homeostasis, ROS formation, cell apoptosis, EC coupling (more efficient in females) cardiotoxicity protection, injury resistance (more efficient in females)	[60,61,62,63,64,65,66]
GPER	structure and function protection (more efficient in males)	[58,59]
VDR	anti-hypertophic/anti-fibrotic activity contractility/relaxation control E2-dependent fatty acid uptake/β-oxidation reduction (females) anti-inflammatory/anti-autophagic activity	[70,71,72,73,74,75,76]

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
