# Peer review of "The Role of Estrogens and Vitamin D in Cardiomyocyte Protection: A Female Perspective"

_biomolecules, 2021, doi:10.3390/biom11121815_

Round 1

Reviewer 1 Report

Women experience a dramatical raise in cardiovascular events after menopause. The decline in estrogens is pointed to as the major responsible trigger for the risk increase in cardiovascular disease (CVD). Indeed, the menopausal transition associates with heart macro-remodeling, that results from a fine-tuned cell micro-remodeling. The remodeling of cardiomyocytes is a biomolecular response to several physiologic and pathologic stimuli, allowing healthy adaptation in normal conditions, or maladaptation and organ architecture detrimental changes in unfavorable environment. Estrogens largely impinge on cardiomyocyte remodeling, but they cannot fully explain the sex-dimorphism in CVD risk. Albeit cell remodeling and adaptation are under multifactorial regulation, vitamin D emerges to exert significant protective effects, controlling some intracellular paths, often shared with estrogen signaling. In postmenopause, the unfavorable association of hyopoestrogenism-D hypovitaminosis may converge in maladaptive remodeling and contribute to CVD risk increase. The aim of this review is to overview the role of estrogens and vitamin D in female cardiac health, speculating on their potential synergistic effect in cardiomyocyte remodeling, an issue not yet fully explored. Further learning the crosstalk between these two steroids in the biomolecular orchestration of cardiac cell fate during adaptation may help the translational approach to future cardioprotective strategies for women health. While this work is interesting, a number of concerns remain.

  1. The authors should make it more clear with regards to the clinical manifestations of sex dimorphism and mechanisms of action behind sex advantage in cardiovascular diseases (consider citing a number of relevant publications PMID 34660569; 25448287; 34650448; 19214173; 15711018 ).
  2. Earlier studies on cardiomyocyte function in estrogen replacement therapy in post-menopause or OVX-mice should be discussed (cardiac remodeling and function).
  3. English expression needs major attention. There are numerous grammatical and stylistic mistakes. Avoid one or two sentences short paragraph. There are too many of these.  
  4. In the present review, vit D and gender seem to be two independent lines. Interplay between these two is not well discussed? Is there sex difference in vitamin D metabolism and levels? The effect of vitamin D on cardiovascular functional regulation between male and female should be discussed.
  5. Interplay between vitamin D and other sex hormones should be briefly mentioned.
  6. The figure is not very effective.

Author Response

We thank the R1 for precious suggestions and comments to ameliorate the paper.

Please find as follows point-by-point responses. Please note that the indicated line number refers to marked text displaying all comments visible.

  1. Comments on clinical manifestations of sex dimorphism and possible mechanisms involved are addressed in paragraph 2, line 67-85 of the revised text; suggested publications are now quoted as number 13, 14, 15, 41 and 69. Please note that references have been re-numbered according to these and other new references quoted in the manuscript, while addressing all reviewers’ requests.
  2. Issues on estrogen replacement therapy and studies in OVX mice are now present and discussed in the revised text, in paragraph 2, line 135-149 and in paragraph 3, line 231-237.
  3. English has been revised from a native speaker; subparagraphs 2.1 and 2.2 have been merged, and, consequently, re-numbered.
  4. We understand this criticism on vitamin D and gender discussed as separate and not as interconnected issues. We appreciate this comment because, on a side, reveals the difficulty emerged from literature to find papers on this specific topic, but, at the same time, it, somehow, strengthens the aim of this review, whose intent is to drive the attention on this point still neglected, albeit potentially impacting on women cardiovascular health. Indeed, while there are several reports onto general vitamin D effects and cardiovascular disease, the standpoint of sex-specific highlights is still missing, particularly considering the effect on cardiomyocyte remodeling. In the revised text, this aspect (albeit already stated in the original text) is additionally mentioned and addressed, paragraph 1, line 50-51, paragraph 4, line 272-273, paragraph 7, line 488-493. Comments on sex-difference in vitamin D basal level as well as comments on vitamin D-induced functional regulation in men and women with cardiovascular diseases are addressed in paragraph 4, line 259-275.
  5. In light of sex-hormone related differences, the interplay of vitamin D with testosterone is mentioned in paragraph 6, line 425-430.
  6. Sorry that figure seems not so effective. Hopefully, the other schemes added upon R2 request may serve to mitigate this defect.

Reviewer 2 Report

Dear Authors,

I send you my comments

1) Methods used to choose the mauscripts are missing, please add

2) please add a table for each section 

3) please add the dosage of vitamin D, Which is the dosage related to the effect of vitamin D on the heart?  

Author Response

Thanks to R2 for the constructive comments. The point-by-point responses are following. Please note that the indicated line number refers to marked text displaying all comments visible.

  1. Methods indeed are not included, as this review is not systematic or a meta-analysis. However, we are glad to share with R2 the methods as following: search was conducted using PubMed and, at less extent, Google Scholar for general terms related to women cardiovascular health (i.e., cardiovascular, disease, incidence, gender, sex, mortality) along with terms related to postfertile life (i.e., menopause, estrogens); the aforementioned term were run with “cardiomyocyte remodeling” or “cardiac remodeling” and “vitamin D”; additional research was run combining the terms estrogens and vitamin D and synergy and cardiac/cardiomyocyte protection. There was no restriction on publication date; non-English papers were excluded.
  2. We added summarizing schemes as new figures 1, 2 and 3 in paragraph 2, 4 and 5, respectively.
  3. Comments on the effective vitamin D dosage related to cardiac effects, as well as serum D level associated with cardiovascular risk are addressed in paragraph 7, line 472-476. Related references have been quoted. Please note that references have been re-numbered according to these and other new references quoted in the manuscript, while addressing all reviewers’ requests.

Reviewer 3 Report

The present review by Crescioli C entitled "The role of estrogens and vitamin D in cardiomyocyte protection: a female perspective" provides an interesting and relevant approach highlighting the important role of vitamin D on cardiovascular function, with special focus on its effect on cardiomyocytes. To date, there are many reviews focused mainly on the role of estrogen on cardiovascular function and its effects when estrogen levels fall after menopause. This review goes even deeper into these aspects while highlighting how vitamin D also plays a key role in cardiovascular protection by describing each of the molecular pathways that are affected by it, as well as the effects of a decrease in its levels. On the other hand, and this is something that is appreciated, during his manuscript Crescioli C conveys the importance of the sexual factor in terms of its effect, making a comparison between what is observed in both sexes, showing the importance of the study in females. 

I found rigorous and correct the methodology of selection and evaluation of data here presented.

Author Response

We are really grateful to R3 for the appreciation of the manuscript, revealing deep understanding of our choice to write on a topic still neglected in literature but with high potentiality to impact female cardiovascular health

Round 2

Reviewer 1 Report

The authors have revised the manuscript and responded well to my previous concerns 

Reviewer 2 Report

Dear authors,

I have read the manuscript and even if I think that it does not reach the leve to be accepted, I have not comments